# Molecular epidemiology of Kaposi sarcoma virus in Spain

Inmaculada Gómez[1], Maria Dolores Pérez-Vázquez[1], David Tarragó[1,2]*

**1** Centro Nacional de Microbiología (CNM), Instituto de Salud Carlos III, Majadahonda, Spain, **2** CIBER Epidemiology and Public Health (CIBERESP), Madrid, Spain

* davtarrago@isciii.es

## Abstract

### Background

Since human herpesvirus 8 (HHV-8) infection may be underestimated and HHV-8 subtype circulation in Spain remains unknown, a molecular epidemiologic study is highly desirable.

### Objectives

This study aimed to analyse HHV-8 subtype diversity and their distribution in Spain.

### Study design

The study included 142 HHV-8 infected patients. A nested PCR was developed in order to permit Sanger sequencing of HHV-8 K1 ORF directly from clinical samples received at the CNM from 2013 to 2021. Phylogenetic characterization was performed.

### Results

Genotypes A and C comprised 55.6% and 42.3% of strains. Regarding subtypes, 25.4% of strains were C3, 19.7% were A3, 14.1% were A5, and C2, A1, A4, C1, A2, C7 were 11.3%, 11.3%, 8.5%, 4.2%, 2.1% and 1.4%, respectively. Subtype E1, E2 and B1 were found in only one patient each (0.7%). The Madrid region accounted for 52.1% of patients and showed a significantly different subtype distribution compared to the others (P = 0.018). Subtypes B1, E1, and E2 were observed to appear sporadically, although overall genotypes A and subtype C3 remained the most frequent and unwavering. Subtype A3 presented the highest diversity as displayed by the highest number of clusters in phylogenetic analysis. Non-significant differences in viral loads between genotypes were found, but significantly higher viral loads in subtype C2 compared to subtype C3 was found, while no significant subtype differences were observed between subtypes within genotype A. Infections with HHV-8 were detected in 94 (66.2%) patients without KS and compared to patients with KS non-significant differences in subtype distribution were found.

### Conclusions

Subtype prevalence and regional distribution followed a similar pattern compared to other western European countries. Our study is the first to report HHV-8 subtypes E1 and E2

**Data Availability Statement:** All relevant data are within the manuscript and its Supporting Information files. Sequence data are available at GenBank: accession numbers ON653051-ON653192.

**Funding:** This work was supported by funds and a grant from Instituto de Salud Carlos III. Project code MPY 1372/2012 and MPY 434/2021. The funders had no role in study design, data collection and analysis, decision to publish, or preparation of the manuscript.

**Competing interests:** The authors have declared that no competing interests exist.

circulating in Europe that might be reflective of migration of population from Caribbean countries. Our study suggests that infection by HHV-8 is underestimated, and wider screening should be recommended for risk groups.

## Introduction

Kaposi sarcoma (KS), which is a type of slow-growing endothelial cell tumour, is characterized by four different clinical and epidemiological settings: classic KS, mainly occurring in elderly men of Mediterranean descent [1]; African-endemic KS [2]; iatrogenic KS, developing in solid organ transplantation recipients [3]; and epidemic or AIDS-associated KS [4], in addition to being associated with Primary Effusion Lymphoma (PEL) and Castleman's Multicentric Disease (CMD) [1].

Prior to the onset of the HIV pandemic, KS was thought to be limited to aged men living in the Mediterranean region. This type of cancer can occur at various sites in the body, but skin lesions are common. Subsequently, in the early 1980s KS emerged as an AIDS-defining tumour. Finally, epidemiological data suggested the involvement of a sexually transmitted agent other than HIV in the development of KS, and this led to the identification of HHV-8 in 1994 by Chang et al. [2]. Initial findings were that HHV-8 was present in cells within KS lesions, but not in surrounding normal tissues.

Currently, the virus has a prevalence of 40% in sub-Saharan Africa; 10% in Mediterranean countries; 2–4% in Northwest Europe, Southeast Asia and the Caribbean; and 5–20% in the United States [3]. However, it is highly likely that these data are underestimated, as the number of publications regarding HHV-8 epidemiology is scarce and incomplete.

The HHV-8 genome is highly conserved between strains (> 90% identity). HHV-8 genotyping is based on the *K1* gene that is expressed during the lytic cycle (*K1*ST) and it is highly variable. Variations are concentrated in the mid-region which is divided into two hypervariable regions (VR1 and VR2) [4], and up to 85% of nucleotide variations in these regions producing amino acid changes [5–7]. Epidemiologists thus consider ORF *K1* as a marker of strain diversity for investigation of HHV-8 epidemiology and transmission. Phylogenetic analysis of the *K1* gene distinguishes seven viral genotypes of HHV-8 (A, B, C, D, E, F, and Z) and up to 22 subtypes or genotype variants that have different distributions across diverse groups based on their geographical and ethnic origin, and also it seems to migrate with the human populations [5, 8–10].

Evolutionary analysis has suggested that genotype B is approximately 100,000 years old in Africa; genotypes D and E appeared approximately 60,000 years ago in the Pacific Islands in Native American populations, respectively; and genotypes A and C appeared approximately 35,000 years ago in Eurasia [11, 12]. Consistent with this scenario, genotypes A and C currently predominate in Europe, the United States, Asian regions and the Middle East, and genotype B is predominant in Africa. The infrequent genotypes D, E and F have only been recorded in the Pacific Islands and Taiwan, in native populations in Latin America, in Uganda and in Brazil, respectively [13, 14]. These genotypes are divided into subtypes A1-A5, B1-B4, C1-C7, D1, D2, E1 and E2 [15]; genotype F has recently been divided into F1 and F2 subtypes [16].

At present, there is no previous study on the detection, prevalence and distribution of the subtypes of this virus in Spain. There is only one study by Marcoval J. et al. [17], which analysed KS cases from Catalonia during the period 2013–2016. They found mainly classical KS

cases and few epidemic and iatrogenic KS cases and none of endemic type. European epidemiology studies of HHV-8 are scarce, and those that deal with it globally, either do so with a non-significant sample size or focus on other European countries such as Italy or Greece where it is supposed to be more prevalent [14, 18].

The aim of this study was to characterize HHV-8 genotypes and subtypes distribution using clinical samples from patients with different pathologies that were submitted to National Center for Microbiology (CNM) from "Instituto de Salud Carlos III" (ISCIII) from 2013 to 2021 in order to determine the molecular epidemiology of the virus in Spain.

## Methods

### Patients and specimens

This retrospective study was performed with positive HHV8 real-time PCR clinical samples from patients with different conditions and living in Spain submitted to the National Center for Microbiology for virological diagnosis of lymphotropic herpes virus including CMV, EBV, HHV6, HHV7 and HHV8, from June 2013 to July 2021. The database was compiled from the clinical and demographic data provided by clinicians of collaborating hospitals. All methods were carried out in accordance with relevant guidelines and UE regulations. All experimental protocols including the use of residual clinical specimens submitted for virological diagnosis and written informed consent from all subjects and/or their legal guardian(s) was approved by the Ethics Committee of the "Instituto de Salud Carlos III" (CEI PI 30_2022-v3).

### DNA extraction, HHV-8 screening, nested PCR design and Sanger sequencing

DNA extraction was performed in the QIAsymphony robot platform (Qiagen). Screening of positive HHV-8 was carried out by quantitative multiplex real time PCR on a Rotor Gene thermal cycler (Qiagen). This assay detects the presence of the following herpesviruses: EBV, CMV, HHV-6, HHV-7 and HHV-8 and it is currently used for diagnostic purposes in the herpesvirus laboratory of the CNM.

A novel nested PCR was developed to amplify ORF *K1*. The first PCR amplification round was performed under the following conditions: 35 cycles of 98˚C for 40 s, 62.6˚C for 10 s, 72˚C for 30s, followed by an elongation step at 72˚C for 5 min. The second round differs only in the annealing temperature, which is 60˚C, instead of 62.6˚C. All reactions were performed using the Platinum SuperFi Green PCR Master Mix kit (Invitrogen). PCR products were purified using the Illustra™ Exostar™ 1-Step Enzymatic PCR and Sequence reaction clean up Kit (GE Healthcare, Germany) and processed for Sanger dideoxy sequencing in a ABI PRISM 3100 sequencer (Applied Biosystems, California, USA). Oligonucleotides designed for nested PCR and sequencing are shown in Table 1.

### DNA sequence processing and amino acid phylogenetic analysis

All sequences were assembled and edited using the Lasergene SeqMan (DNASTAR; INC) software and aligned with reference sequence (GenBank KT215109.1). Once translated they were aligned using MEGA-X software with 101 ORF-K1 GenBank available sequences of most of subtypes. GenBank sequences and those obtained in the present study were analysed and used to construct a phylogenetic tree using IQtree software. Pairwise genetic distance among K1 amino acid sequences were calculated with 1000-bootstrap resampling, using JTT model of substitutions and a gamma distribution with 4 parameters. Subtype ORF-K1 reference sequences were the following: For subtype A1, AF133038, FJ884626 and KT215151; for

**Table 1. Primers used for nested PCR and sequencing.**

| | Primer | Sequence (5′- 3′) | Nucleotide positions[a] |
|---|---|---|---|
| **K1 VHH-8 1 (1043 bp)** | Primer forward | TCAGACCTTGTTGGACATCC | 73–92 |
| | Primer reverse | GCCATGCTGTAAGTAGCACG | 1097–1116 |
| **K1 VHH-8 2 (824bp)** | Primer forward | TTCCTGTATGTTGTCTGCAG | 108–127 |
| | Primer reverse | CACTGGTTGCGTATAGTCTT | 913–932 |
| **K1 VHH-8 Sequencing** | Forward_2 | TCTGCCCTGGAGTGATTTCAACG | 184–206 |
| | Forward_3 | GTTACCGTTTGGCATCTACCA | 474–494 |
| | Forward_4 | GCAACCGCACCTACTCTATTTGT | 693–715 |
| | Reverse_1 | GGCTGTATCGATGCCCAGATTGT | 338–360 |
| | Reverse_2 | AGACGACAGCCCGTTAGAACAAG | 571–593 |
| | **Reverse_3** | GGCACTGTTTTGTTTGAGTCACGTTG | **843–868** |

[a] according to NCBI Reference Sequence of HHV-8: NC_009333.1

subtype A2, U75698, AF178807, KT215108 and AF130305; for A3, AF130287, AF178799, AF278829, AY756111, AY756112, FJ884608, GU097420 and KU950281; for A4, AF130299, AF133039 and GU097430; for A5, AF130282, AF130284, AF130289, AF178823, AY377993 and AY378004; for B1, AF133040, AF178801, KT215112, KT215113, MT510658, MT510662 and MT510670; for B2, AF178792, AF178804, AY042947, AY953877, MH632203 and MH632211; for B3, AY042940, AY042941, FJ884617, FJ884619, FJ884620, KF781670, KF781674, KF781682 and MT510669; for B4, DQ309732, DQ309739, DQ309742, DQ309744, DQ309748, DQ309749, DQ309752 and DQ309762; for C1, AF130267, AF130273, DQ394055 and GU097423; for C2, AF130304, DQ394048, FJ853367, FJ853368, FJ853369, FJ853380, FJ853381, FJ853385 and FJ853387; for C3, DQ394035, DQ394036, DQ394059, FJ853379 and U93872; for C4, SKS1 (from Zong et al., 1999) for C5 SKS9 (from Zong et al., 1999 [6]), AM423130 and AM423136; for C6, IKS7 (from Zong et al., 2002 [12]) and AY204649; for C7 DQ394056, DQ394061 and DQ394062; for D1, AF133043; for D2, AF133044, for E AY329027, AY850983, EF153264, FJ986135, FJ986136 and FJ986137; for E1, AF220292 and AF220293; for E2, AY329028 and AY940426; for F1, AY953882 and QJF74392; for F2, AF178810, QJF74390, QJF74391, QJF74395, QJF74418 and QJF74419.

## Statistical analysis

A descriptive analysis of the qualitative variables was performed. Frequency distribution of variables was analysed. The Pearson's Chi-square test or, when appropriate, Fisher's exact test for trend was used to assess statistical differences in the proportion of genotypes and subtypes and their geographical origin. Analysis was performed using SPSS version 28.0 software (SPSS, Chicago, IL). Differences were considered statistically significant when p-values were $\leq 0.05$. One-Way ANOVA was used to compare the viral load of HHV8 between the subtypes within A and C genotypes, followed by Tukey's test for multiple comparisons between means (SD), 95% CI and p-value $\leq 0.05$.

## Results

### Patients and clinical samples

A total of 142 positive-PCR HHV-8 clinical samples from 142 patients were sequenced and analysed. Regarding the descriptive analysis of the clinical results, 48 (33.8%) patients were diagnosed with KS, 15 (10.6%) patients were diagnosed with CMD, and 5 (3.5%) patients were

**Table 2. Baseline characteristics of the 142 patients included in the study.**

| Variables | Total (N = 142) | | Kaposi Sarcoma (N = 48) | |
|---|---|---|---|---|
| | n | % | n | % |
| *Age* | | | | |
| <50 years | 77 | 54.2 | 26 | 54.2 |
| ≥50 years | 63 | 44.4 | 20 | 41.7 |
| Unknown | 2 | 1.4 | 2 | 1.4 |
| *Diagnosis related to HHV-8 infection* | | | | |
| HIV+ | 51 | 35.9 | 27 | 56.3 |
| Cryptococcosis and HIV+ | 1 | 0.7 | 0 | 0.0 |
| AIDS | 10 | 7.0 | 4 | 8.3 |
| CMD | 15 | 10.6 | 8 | 16.7 |
| PEL | 5 | 3.5 | 1 | 2.1 |
| *Other Infections* | | | | |
| CMV | 3 | 2.1 | 2 | 4.2 |
| CMV and EBV | 1 | 0.7 | 1 | 2.1 |
| HHV-6 | 1 | 0.7 | 0 | 0.0 |
| Non-HHV-8 classical associated diseases | 74 | 52.1 | 0 | 0.0 |
| Non-expected HHV-8 infection | 61 | 43.0 | 0 | 0.0 |
| Not previous diagnosis | 32 | 22.5 | 0 | 0.0 |
| *Region* | | | | |
| Andalusia | 12 | 8.5 | 1 | 2.1 |
| Aragon | 2 | 1.4 | 1 | 2.1 |
| Asturias | 1 | 0.7 | 0 | 0 |
| Canary Islands | 1 | 0.7 | 1 | 2.1 |
| Cantabria | 4 | 2.8 | 2 | 4.2 |
| Castilla-Leon | 7 | 4.9 | 1 | 2.1 |
| Castilla-La Mancha | 3 | 2.1 | 0 | 0.0 |
| Catalonia | 3 | 2.1 | 2 | 4.2 |
| Valencia | 6 | 4.2 | 2 | 4.2 |
| Extremadura | 1 | 0.7 | 1 | 2.1 |
| Galicia | 10 | 7.0 | 5 | 10.4 |
| Balearic Islands | 2 | 1.4 | 2 | 4.2 |
| Madrid | 74 | 52.1 | 20 | 41.7 |
| Navarre | 2 | 1.4 | 2 | 4.2 |
| Basque Country | 14 | 9.9 | 8 | 16.7 |

Note: VIH, Human immunodeficiency virus; AIDS, Acquired Immune Deficiency Syndrome; EBV, Epstein–Barr virus; CMV, Cytomegalovirus; CMD, Castleman disease; PEL, Primary effusion lymphoma.

diagnosed with PEL. Remaining 94 HHV-8 infected patients (66.2%) were not diagnosed with KS or any of the HHV-8 associated disease.

There was no reported epidemiological relationship between patients. Table 2 summarizes characteristics of the patients included in the study.

Many cases included in this study were epidemic type as 51 patients were HIV infected, 23 of them had only KS symptoms, 6 of them presented AIDS but non-KS symptoms and 4 of them presented both. The remaining 18 HIV infected patients were positive by RT-PCR HHV-8, but they did not present KS sympthoms and they had not been classified as AIDS patients. In 63 patients ≥50 years old HHV-8 was detected by RT-PCR, 20 of them previously

diagnosed of classical KS. Regarding iatrogenic cases, 9 patients who had recently received a transplant were positive by RT-PCR HHV-8, 4 of them presented KS symphoms and 5 of them do not. Regarding probable African endemic cases, 8 positive RT-PCR HHV-8 patients were from endemic KS countries although only 3 of them had developed KS symptoms.

Clinical samples were the following: 80 (56.3%) were blood, 24 (16.9%) plasma, 7 (4.9%) serum, 5 (3.5%) pleural fluid, 4 (2.8%) Broncho alveolar wash, 4 (2.8%) skin biopsy, 2 (1.4%) bone marrow, 2 (1.4%) gastric biopsy, 1 (0.7%) Broncho alveolar aspirate, 1 (0.7%) unspecific biopsy, 1 (0.7%) colon biopsy, 1 (0.7%) bowel biopsy, 1 (0.7%) oesophagus biopsy, 1 (0.7%) lung biopsy, 1 (0.7%) lymph node biopsy 1 (0.7%) lesion exudate, 1 (0.7%) oral exudate, 1 (0.7%) ascites, 1 (0.7%), 1 (0.7%) pericardial fluid, 1 (0.7%) saliva and 1 (0.7%) bloodspot on paper.

## Distribution of HHV-8 subtypes and phylogenetic analysis of ORF-K1

We performed a phylogenetic analysis of 101 ORF-K1 amino acid sequences available in Gen-Bank and the 142 (P1-P142) sequences obtained in the current study which were uploaded to GenBank with accession numbers ON653051-ON653192 (rooted phylogenetic tree at mid-point in Fig 1A and unrooted tree in Fig 1B). Among 142 HHV-8 infected patients, 21 different subtypes were found. Genotype A was the most prevalent with 79 cases (55.6%), followed by genotype C with 60 cases (42.1%). Genotype E and B were found in 2 patients (1.4%) and 1 patient (0.7%) respectively.

Regarding subtypes, the most prevalent was subtype C3 (N = 36, 25.4%) followed by A3 (N = 28, 19.7%), A5 (N = 20, 14.1%), C2 (N = 16, 11.3%); A1 (N = 16, 11.3%), A4 (N = 12, 8.5%), C1 (N = 6, 4.2%), A2 (N = 3, 2.1%), C7 (N = 2, 1.4%). Subtype E1, E2 and B1 were 1 each one (0.7%). Co-infection with more than one subtype was not found. F1 and F2 subtypes were not found.

Most of the subtypes were close to the corresponding reference subtype strains in the same cluster. Interestingly, most subtype A3 sequences (17 of them) were only closely related to a sequence first described by Meng et al. 1999, reclassified as A3 by Matteoli et al. 2012 (AF278829). Another cluster of A3 sequences (P77, P78 and P91) were not closely related to any K1 sequence previously described and they contained a specific polymorphism in the K1 hypervariable region ([9]RQ.F.VTTL[17]).

Most KS patients had C3 and A3 subtypes 13 (27.1%) each, followed by the A5 subtype with 8 (16.7%) patients. Concerning KS types, 3 (6.25%) were classified as endemic KS patients with subtype A5; 8 (16.6%) were epidemic KS patients wth A3 subtype, 7 (14.6%) were classical KS patients and 2 (50%) iatrogenic KS patients both with C3 subtype. For 94 non-KS patients, C3 subtype was the most prevalent with 23 (24.5%) patients, followed by A3 subtype with 15 (16.0%) patients. Regarding associated KS pathologies (PEL and CMD), C3 was the most prevalent subtype with 8 (53.3%) and 2 (40%) patients with CMD and PEL, respectively.

Concerning HHV-8 Ct values of RT-PCR, taken as an indirect measure of viral load, no significant differences were found between genotype A and C (One-Way ANOVA P = 0.087 and Tukey's test P = 0.125). But a significantly higher viral load was observed in subtype C2 compared to C3 (One-Way ANOVA P = 0.005 and Tukey's test, between C2 and C3, P = 0.008). Non-significant differences were observed among A subtypes (One-Way ANOVA P = 0.131 and Tukey's test P = 0.170).

## Distribution of genotypes and subtypes in the timeline 2013–2021

When distribution of genotypes and subtypes over time was analysed, genotype A was the most prevalent during the 9-years of study. Except for 2016 and 2018 when the number of

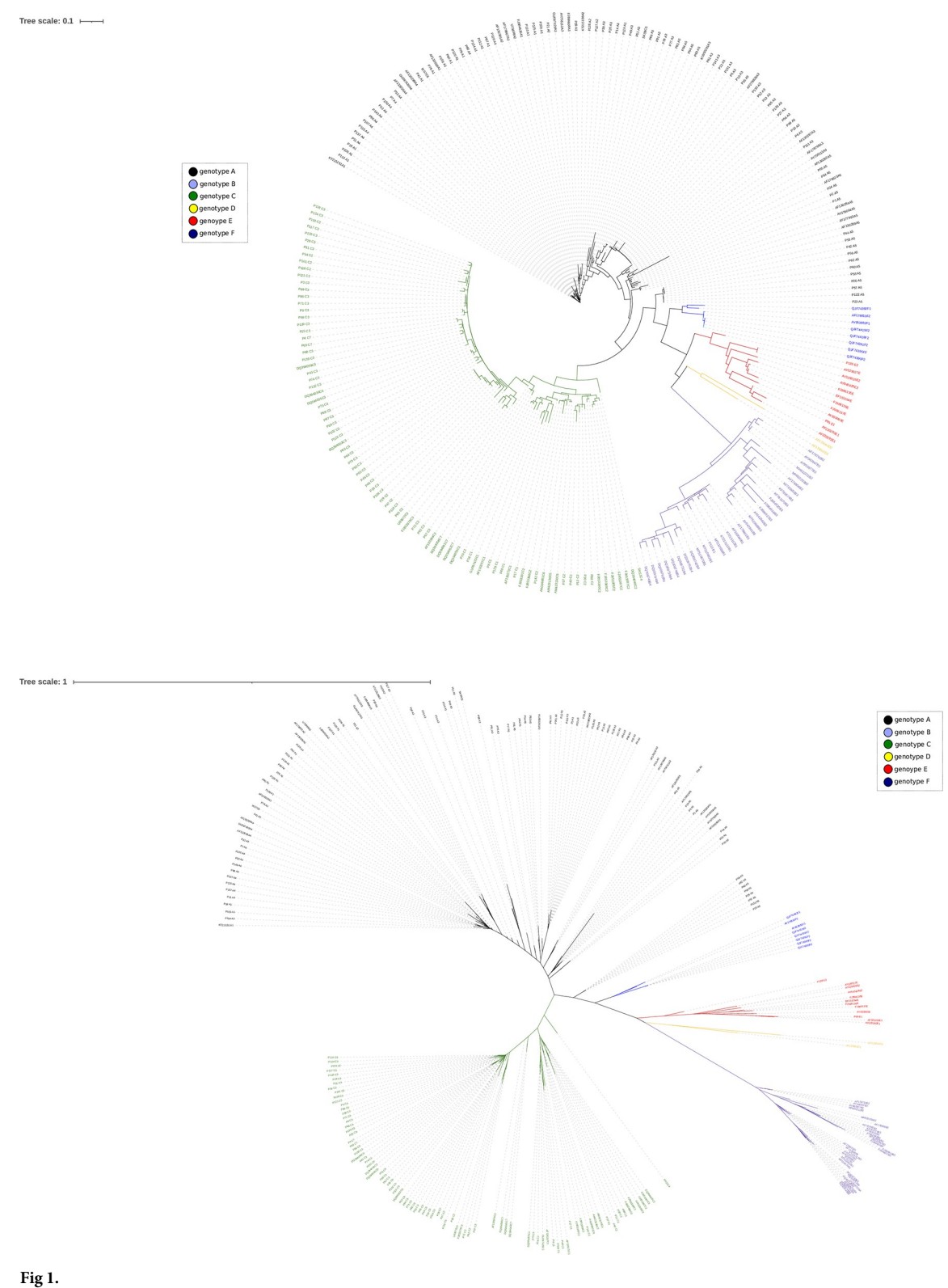

**Fig 1.**

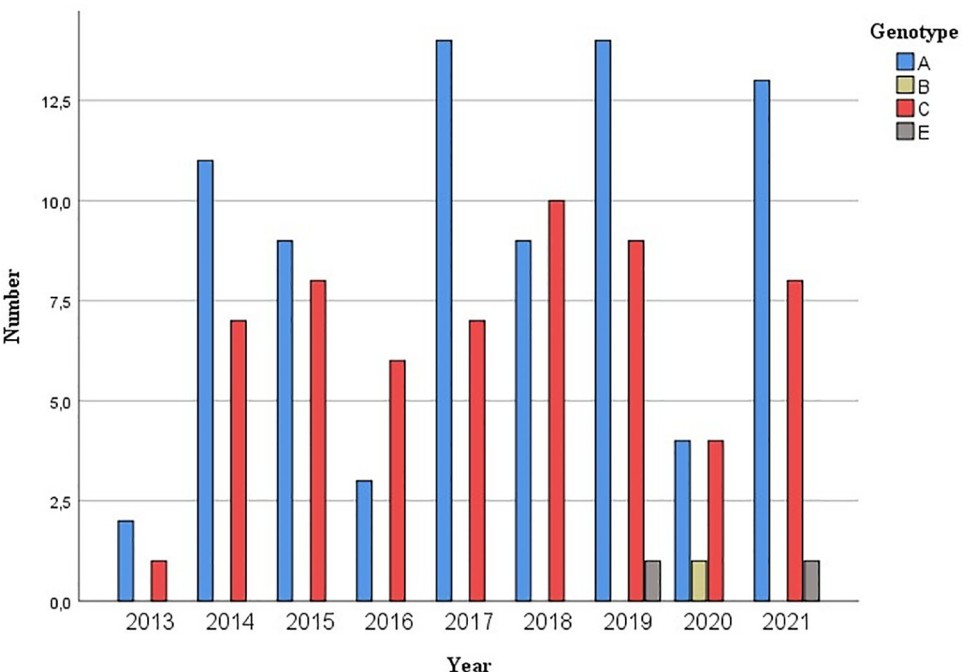

**Fig 2. Graphical representation of the temporal distribution of the different HHV-8 genotypes.**

patients with genotype C was the highest and 2020 when genotype A and C were the same (Fig 2). If subtypes were analysed, subtype C3 was always the most prevalent over the period (Fig 3). A limitation for this analysis was that 2013, 2016 and 2020 were underrepresented.

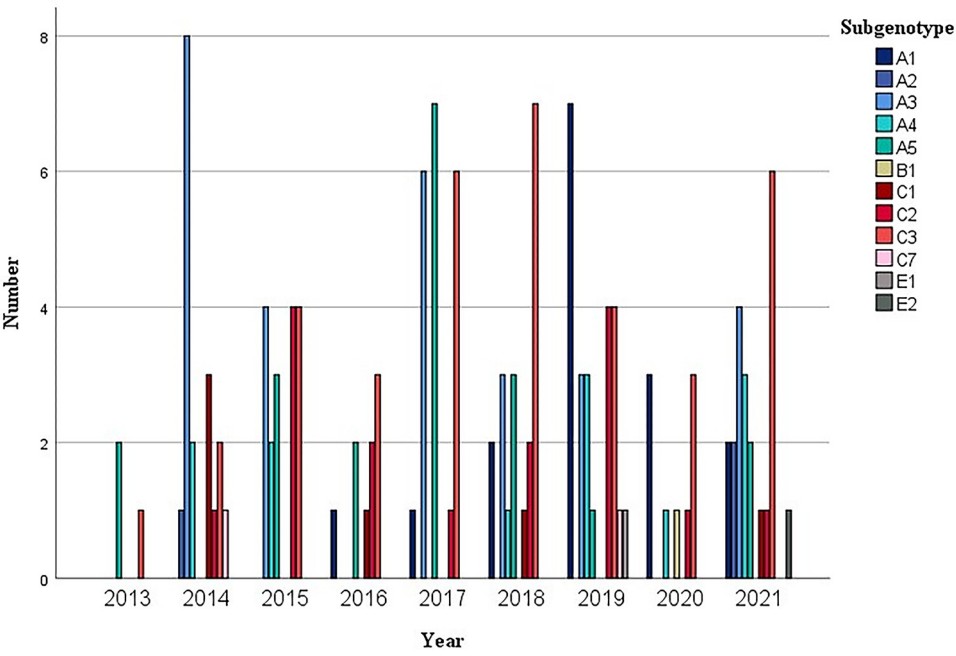

**Fig 3. Graphical representation of the temporal distribution of the different HHV-8 subtypes.**

## Distribution of genotypes and subtypes regarding Spanish regions

Geographic distribution of genotypes and subtypes was analysed for most of the Spanish regions. The Madrid region provided the highest number of patients 74 (52.1%) out of 142 total patients, followed by Basque Country with 14 patients (9.9%). Andalusia and Galicia with 12 (8.5%) and 10 (7.0%) patients, respectively. A smaller number of patients accounted for the rest of regions compared to the above-mentioned.

The urban area of Madrid accounted for most of patients with non-European origin: 25 (33.8%) Europeans; 16 (21.6%) Non-Europeans. Therefore, significant differences in genotype and subtype distribution were found in Madrid (P = 0.001 and P = 0.018, respectively) compared to the rest of the country. Non-significant differences were found in genotypes and subtypes distribution (P = 0.803 y P = 0.188, respectively) between northern vs southern regions, excluding Madrid region as out-of-the-ordinary group. C3 subtype was observed in all regions except Asturias, Galicia, Castilla-La Mancha and Navarra. Subtypes with lower presence in Spain (0.7%) were B1, E1 and E2, which were all found in Madrid region.

## Discussion

To date, this is the first study aimed at determining the distribution of HHV-8 subtypes in Spain providing a basis for future epidemiological studies of HHV-8 in this country and the European Union. Typing and phylogenetic analysis of HHV-8 were performed on 142 patients living in Spain with different clinical conditions from 15 of 17 Spanish regions for 9 years, which represents a reliable picture of HHV-8 subtypes in circulation. The molecular classification of HHV-8 strains was based on ORF *K1* gen, which is commonly acknowledged in molecular studies focused on determination of the origin, genetic evolution, transmissibility, disease associations and epidemiology of the virus [19–21].

In the current study, molecular analysis indicated that subtypes C3 and A3 were the predominant subtypes in accordance with previously described in other European countries [4, 14, 16, 22].

The 142 strains were subtyped accordingly to previous subtyped strains available at GenBank, as could be observed at phylogenetic tree (Fig 1A and 1B). Subtype A3 was the most diverse as determined by the highest number of different genetic clusters, some of them separated into novel branches from most previous subtype A3 GenBank strains. Interestingly, a crowded cluster of 17 subtype A3 strains were only related to a sequence first described by Meng et al. 1999 [23], reclassified as A3 by Matteoli et al. 2012 [24] and another three A3 strains shared a specific polymorphism not previously described. Subtype A5 were found in 20 (14.1%) of patients, 8 of them were reported of African origin (Nigeria, Cameroon, Equatorial Guinea and Morocco) with endemic KS but only 3 were reported as KS. Three cases of subtype A5 were from patients from South America origin and the 9 were from European origin (Spain). These data are consistent with previous studies that determined the presence of some subtypes A and C [14, 19] and, in particular, subtype A5 in African countries [4, 21]. However, the presence of 9 cases of A5 from patients of European origin (Spaniards) were remarkable. Subtypes with the lowest prevalence such as E1, E2 and B1 were from patients living in Spain but born in Cuba, Romania and Mexico, respectively: These subtypes have been described to be widespread in Africa and South/Central America [14]. To our knowledge this is the first case reported in Europe of genotype E. Subtypes E1 and E2 have been previously described in Amerindians [25] and it was also reported in a Cuban study [26]. Consequently, it was consistent that our E1 strain (P95) and those of the Cuban study were found in the same cluster as it has been shown in Fig 1A and 1B.

Recently, in a French study carried out by Jary et al. [15] a novel F2 subtype with a relevant prevalence in Caucasian MSM is described. Sexual behaviours were not recorded in the current study and F genotype were not found. However, we did not rule out the existence of F2 subtype cases in MSM living in Spain.

Regarding the prevalence of subtypes over time in Spain, there were no previous studies to which to compare. A background of infection with subtypes C3 and A3 seems to be constant although other minor subtypes appeared sporadically related to detection of infection from people from non-European geographic areas. However, it should be noted that 2013, 2014 and 2020 years are underrepresented, the latter due to pandemic COVID.

Although our study design is unable to yield incidence data, we did not observed differences in incidence or subtype distribution as previously described in the incidence of classical KS in other European Mediterranean regions like Italy [22]. Surprisingly, an important number of HHV-8 infected patients (94 patients, 66.2%), HHV-8 were detected with no previous diagnosis of KS. Moreover, most of them (61 patients, 43.0%) were not expected by clinicians to be HHV-8 infected and 32 patients (22.5%) were undiagnosed. It could be explained because of the long course of infection and disease which results in a late development of clinical symptoms. This finding highlights the need of HHV-8 screening overall in high-risk groups to anticipate interventions before development of symptoms of disease. A study by Marcoval J et al. [17] analysed 191 KS cases from Catalonia in the period 2013–2016 and they found that the number of annual cases of classic KS and immunosuppression-associated KS is increasing.

Epidemiological studies showed a geographical distribution of HHV-8 infection and subtypes that seemed to be accompanied by social factors in certain geographical areas which was not applicable amongst different Spanish regions. Risk factors associated with KS prevalence and other HHV8-related diseases were age and an immunocompromised status caused by drugs, the host's genetic factors, viral infections like HIV [27] and parasite infections like malaria [28]. Studies from Uganda and Gabon found an association between malaria transmission and HHV-8 prevalence [25]. In this sense, further studies are needed to stablish this association in ancient malaria endemic areas from the South of Europe. Although malaria was declared non-endemic in Spain in 1964, more than 10,000 cases have been reported with a steady increase trend since 2008, mostly found in travellers and migrants. Nowadays, it is the most frequently imported disease in this country [29].

Significant subtype differences were observed in viral load, in particular, subtype C2 showed higher Ct values than subtype C3 while non-significant differences were observed amongst. Non-significant differences were previously described in other studies [16, 19], although other authors found differences according to cell tropism [24]. It was also previously suggested that an association between HHV-8 subtypes and different tumour and pathogenic potentials [9, 30]. Although it exceeded the scope of the current study, other studies have confirmed this association [5, 16, 30–33]. Even so, there is still controversy about this matter [26, 34].

In conclusions, despite being a virus discovered a decades ago, intriguing matters that deserve attention must be further studied as the effect of spread of African and South-Central American subtypes in Europe together with immunosuppression and other host cofactors, which may change HHV-8 epidemiology, pathogenesis and clinical manifestations of disease. The finding that HHV-8 infection is underestimated supports the need to increase screening and genotyping of HHV-8 overall in high-risk groups.

## Supporting information

**S1 File. Alignment of subtype sequences from clinical samples and their corresponding reference sequences.**
(DOCX)

**S1 Table. Detailed information for each patient regarding date of request, age, native country hospital of origin and subtype.**
(DOCX)

**S2 Table. Number and proportion of detected subtypes of HHV-8 in the different regions of Spain during the period 2013–2021.**
(DOCX)

## Author Contributions

**Conceptualization:** David Tarragó.

**Data curation:** Inmaculada Gómez, Maria Dolores Pérez-Vázquez.

**Formal analysis:** Inmaculada Gómez, Maria Dolores Pérez-Vázquez.

**Funding acquisition:** David Tarragó.

**Investigation:** David Tarragó.

**Methodology:** Inmaculada Gómez.

**Validation:** David Tarragó.

**Writing – original draft:** Inmaculada Gómez.

**Writing – review & editing:** David Tarragó.

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
