## [Decision Letter · Decision Letter 0]

23 Feb 2022

PONE-D-22-01291MOLECULAR EPIDEMIOLOGY OF KAPOSI SARCOMA VIRUS IN SPAINPLOS ONE

Dear Dr. Tarragó,

Thank you for submitting your manuscript to PLOS ONE. After careful consideration, we feel that it has merit but does not fully meet PLOS ONE’s publication criteria as it currently stands. Therefore, we invite you to submit a revised version of the manuscript that addresses the points raised during the review process.

We look forward to receiving your revised manuscript.

Kind regards,

Dirk P. Dittmer, Ph.D.

Academic Editor

PLOS ONE

Journal Requirements:

4. Thank you for stating the following in the Funding Section of your manuscript: 

"This work was supported by funds and a grant from Instituto de Salud Carlos III. Project code MPY1372/12 and AESI2021 PI21CIII/00011 .The funder had no role in study design, data collection and analysis, decision to publish, or preparation of the manuscript."

We note that you have provided funding information. However, funding information should not appear in the Funding section or other areas of your manuscript. We will only publish funding information present in the Funding Statement section of the online submission form. 

"DT. This work was supported by funds and a grant from Instituto de Salud Carlos III. Project code MPY1372/12 and AESI2021 PI21CIII/00011. The funders had no role in study design, data collection and analysis, decision to publish, or preparation of the manuscript."

7. We note that Figure 3 in your submission contain map/satellite images which may be copyrighted. All PLOS content is published under the Creative Commons Attribution License (CC BY 4.0), which means that the manuscript, images, and Supporting Information files will be freely available online, and any third party is permitted to access, download, copy, distribute, and use these materials in any way, even commercially, with proper attribution. For these reasons, we cannot publish previously copyrighted maps or satellite images created using proprietary data, such as Google software (Google Maps, Street View, and Earth). For more information, see our copyright guidelines: http://journals.plos.org/plosone/s/licenses-and-copyright.

a. You may seek permission from the original copyright holder of Figure 3 to publish the content specifically under the CC BY 4.0 license.  

Reviewers' comments:

Reviewer's Responses to Questions

**Comments to the Author**

1. Is the manuscript technically sound, and do the data support the conclusions?

Reviewer #1: Partly

2. Has the statistical analysis been performed appropriately and rigorously? 

Reviewer #1: No

3. Have the authors made all data underlying the findings in their manuscript fully available?

Reviewer #1: No

4. Is the manuscript presented in an intelligible fashion and written in standard English?

Reviewer #1: No

5. Review Comments to the Author

Reviewer #1: Review of manuscript “Molecular Epidemiology of Kaposi sarcoma virus in Spain” by Gómez I. et al.

Summary

The study describes the analysis of 142 KSHV K1 subtypes, circulating in Spain between 2013-2021. As pointed out by the authors, not many KSHV genotyping studies have been published from Spain, so this is a welcome report for KSHV researchers. The majority of KSHV K1 subtypes identified were A and C as would be expected in Europe, however, rare subtypes E and F were also found. In particular, the E1 subtype discovery is interesting as this KSHV K1 subtype has not been reported in Europe previously to my knowledge.

The authors also report a variant of the A3 subtype that has a unique series of eight polymorphisms. In addition, variation geographically in K1 subtype distribution was not noted in Spain as has been reported for other European countries like Italy.

Major Concerns

The primary major concern is the lack of phylogenetic analysis and a phylogenetic tree. An evolutionary analysis of the study sequences visualized as a phylogenetic tree, for instance using IQTree or BEAST software, would be a necessary addition to the paper. While the amino acid alignments are informative, the evolutionary relationships between the individual subtypes are lost. As mentioned, to date the E1 subtype has only been found in indigenous peoples from South America. More recently, studies from Cuba have identified an E2 subtype that is similar to but distinct from the original E. A phylogenetic tree with both E1 and E2 references would help inform readers as to the evolutionary relationship of the E subtype from the current study, particularly as ethnicity information is not known for the individual. A similar argument could be made for the three F subtypes. It would be interesting to understand the relationship of the current study F subtypes with the F2 reported from French studies. Lastly, the A3 polymorphisms described in 16 of 19 sequences would be best visualized in a phylogenetic tree analysis to better understand the evolutionary distances within the A3 branch as well as the contribution the newly described sequences provide to the KSHV A genotype overall.

The authors should highlight further in the discussion that they are reporting the first subtype E KSHV in Europe and the additional subtype F in relation to the recent French studies.

In general many of the references are older and more recent highly relevant references such as the JAry el paper describing sub=type F in France should be added.

A major concern for the paper is that there are frequent misspellings that detracts from the message. These are easily addressable but would need to be corrected prior to publication.

Minor Concerns

It wasn’t clear what is meant by the statement “Only subgenotype C2 showed significant higher viral loads than subgenotype C3”. Does this mean that C3 and C2 subtype samples were observed to have higher viral loads than the other KSHV K1 subtypes found in the study, for instance A3 and A5, but that C2 displayed the highest VL overall? This is later clarified in the results section, lines 241-245, but as a summary point in the abstract needs clarification.

KSHV K1 A5, B1, C3 etc. are more commonly referred to as “subtypes” in the literature. Don’t necessarily need to replace “sub-genotype” description with “subtype”, just an observation.

Jary et al. recently defined the F2 subtype as distinct from F subtypes commonly found in Africa, now referred to as F1. This can be added to the introduction in line 87.

In line 147, it was noted that only three A3 subtypes were used for comparison in the sequence alignments. Did BLAST searches of the newly described A3 indicate homology to any other published KSHV K1 subtype references?

In lines 164-166, it was noted that only one reference was used for comparison for the rare KSHV K1 subtypes. For some, like C6, there are few available, but the sequence comparisons would be strengthened with the addition of more references. The E2 reference sited from the Biggar et al paper would be considered an E1 due to the publication of E subtypes from Cuba (Kouri,V et al AIDS 2007 and Virology 2012). It would be of interest to compare the current study sequence to the Cuban sequences, as previously mentioned, in order to confirm or clarify the E subtype found in Spain. In addition, F2 subtype references should be added from Lacoste,V. Virology 2000 or Jary et al. J. Inf. Diseases 2020 publications.

In lines 225-227, there is some confusion about why the author’s feel that subtype C3 is the most prevalent. The sentence “This could be explained by the fact that the number of subgenotypes obtained for genotype A (A1-A5) was higher than in genotype C (C1, C2, C3 and C5)” doesn’t seem to support why C3 is found more frequently.

In section 3.3, could the author’s comment if they think that the availability of samples collected year to year could affect the observations made concerning frequency of subtypes observed?

In sections 3.4 and 3.5, some of the observations in certain categories are based upon small sample sizes which should be discussed as a limitation.

In the discussion, it was noted that the frequency of KSHV detection did not vary significantly north to south in Spain as has been reported in Italy. Studies in Italy, as well as Uganda, have found associations between malaria transmission and KSHV prevalence. While the authors admit that the study wasn’t designed to address this question, perhaps they could comment on malaria incidence as a risk factor for KSHV in Spain as well as list the social factors referred to in lines 343-345.

In the supplemental alignment tables, not sure what is meant by alignment of “problem” samples? The reference used for alignment is a KSHV A2 subtype. Alignment of sequences that are not A2 would have polymorphisms. Suggest “problem” be removed from the titles or discussed within the manuscript for clarification.

No mention of data availability was found in the manuscript. Will the sequences be submitted to GenBank or some other publicly assessable resource?

Concluding comment

Publications of KSHV sequences from an understudied area would be of interest to the KSHV community. As KSHV K1 subtypes reflect ethnic and geographical differences, the diverse distribution of subtypes noted in the current study highlight that urban centers may have several circulating strains which can vary over time as communities change.

6. PLOS authors have the option to publish the peer review history of their article (what does this mean?). If published, this will include your full peer review and any attached files.

Reviewer #1: No

---

## [Author Response · Author response to Decision Letter 0]

12 Aug 2022

PONE-D-22-01291

MOLECULAR EPIDEMIOLOGY OF KAPOSI SARCOMA VIRUS IN SPAIN

PLOS ONE

Dear Dr. Tarragó,

Thank you for submitting your manuscript to PLOS ONE. After careful consideration, we feel that it has merit but does not fully meet PLOS ONE’s publication criteria as it currently stands. Therefore, we invite you to submit a revised version of the manuscript that addresses the points raised during the review process.

We look forward to receiving your revised manuscript.

Kind regards,

Dirk P. Dittmer, Ph.D.

Academic Editor

PLOS ONE

Journal Requirements:

https://journals.plos.org/plosone/s/file?id=wjVg/PLOSOne_formatting_sample_main_body.pdf and https://journals.plos.org/plosone/s/file?id=ba62/PLOSOne_formatting_sample_title_authors_affiliations.pdf. This requirement has been addressed.

For additional information about PLOS ONE ethical requirements for human subjects research, please refer to http://journals.plos.org/plosone/s/submission-guidelines#loc-human-subjects-research. This requirement has been addressed.

This mistake has been addressed.

4. Thank you for stating the following in the Funding Section of your manuscript: 

"This work was supported by funds and a grant from Instituto de Salud Carlos III. Project code MPY1372/12 and AESI2021 PI21CIII/00011 .The funder had no role in study design, data collection and analysis, decision to publish, or preparation of the manuscript."

We note that you have provided funding information. However, funding information should not appear in the Funding section or other areas of your manuscript. We will only publish funding information present in the Funding Statement section of the online submission form. 

"DT. This work was supported by funds and a grant from Instituto de Salud Carlos III. Project code MPY1372/12 and AESI2021 PI21CIII/00011. The funders had no role in study design, data collection and analysis, decision to publish, or preparation of the manuscript."

This requirement has been addressed

7. We note that Figure 3 in your submission contain map/satellite images which may be copyrighted. All PLOS content is published under the Creative Commons Attribution License (CC BY 4.0), which means that the manuscript, images, and Supporting Information files will be freely available online, and any third party is permitted to access, download, copy, distribute, and use these materials in any way, even commercially, with proper attribution. For these reasons, we cannot publish previously copyrighted maps or satellite images created using proprietary data, such as Google software (Google Maps, Street View, and Earth). For more information, see our copyright guidelines: http://journals.plos.org/plosone/s/licenses-and-copyright.

a. You may seek permission from the original copyright holder of Figure 3 to publish the content specifically under the CC BY 4.0 license. 

This requirement has been addressed.

This requirement has been addressed.

Reviewers' comments:

Reviewer's Responses to Questions

Comments to the Author

1. Is the manuscript technically sound, and do the data support the conclusions?

Reviewer #1: Partly

The manuscript has been rewritten acordingly with the new data obtained from phylogenetic analysis

2. Has the statistical analysis been performed appropriately and rigorously? 

Reviewer #1: No

Statistical analysis has been performed taking into consideration a previous study by Oliveira et al which analysed variables such as viral load and other variables presented in the currrent study. A. de Oliveira Lopes, N. Spitz, C.R. de S. Reis, V.S. de Paula, Update of the global distribution of human gammaherpesvirus 8 genotypes, Sci. Reports 2021 111. 11 (2021) 1-9. https://doi.org/10.1038/s41598-021-87038-9

3. Have the authors made all data underlying the findings in their manuscript fully available?

Reviewer #1: No

All sequence data have been uploaded to GenBank. Therefore we obtained 142 accesion numbers corresponding to 142 sequences yielded in the current study.

4. Is the manuscript presented in an intelligible fashion and written in standard English?

Reviewer #1: No

The manuscript has been corrected by an American scientis working in the microbiology field.

5. Review Comments to the Author

Reviewer #1: Review of manuscript “Molecular Epidemiology of Kaposi sarcoma virus in Spain” by Gómez I. et al.

Summary

The study describes the analysis of 142 KSHV K1 subtypes, circulating in Spain between 2013-2021. As pointed out by the authors, not many KSHV genotyping studies have been published from Spain, so this is a welcome report for KSHV researchers. The majority of KSHV K1 subtypes identified were A and C as would be expected in Europe, however, rare subtypes E and F were also found. In particular, the E1 subtype discovery is interesting as this KSHV K1 subtype has not been reported in Europe previously to my knowledge.

The authors also report a variant of the A3 subtype that has a unique series of eight polymorphisms. In addition, variation geographically in K1 subtype distribution was not noted in Spain as has been reported for other European countries like Italy.

Major Concerns

The primary major concern is the lack of phylogenetic analysis and a phylogenetic tree. An evolutionary analysis of the study sequences visualized as a phylogenetic tree, for instance using IQTree or BEAST software, would be a necessary addition to the paper. While the amino acid alignments are informative, the evolutionary relationships between the individual subtypes are lost. As mentioned, to date the E1 subtype has only been found in indigenous peoples from South America. More recently, studies from Cuba have identified an E2 subtype that is similar to but distinct from the original E. A phylogenetic tree with both E1 and E2 references would help inform readers as to the evolutionary relationship of the E subtype from the current study, particularly as ethnicity information is not known for the individual. A similar argument could be made for the three F subtypes. It would be interesting to understand the relationship of the current study F subtypes with the F2 reported from French studies. Lastly, the A3 polymorphisms described in 16 of 19 sequences would be best visualized in a phylogenetic tree analysis to better understand the evolutionary distances within the A3 branch as well as the contribution the newly described sequences provide to the KSHV A genotype overall.

In results section a new subheading has been added which includes phylogenetic trees with all our K1 sequences (142) and all mentioned above by the reviewer (101 sequences). Moreover, all this suggestions have been added to discussion

The authors should highlight further in the discussion that they are reporting the first subtype E KSHV in Europe and the additional subtype F in relation to the recent French studies.

This point has been addressed.

In general many of the references are older and more recent highly relevant references such as the JAry el paper describing sub=type F in France should be added.

New references have been added including the reference cited by the reviewer and others concerning E subtypes found in Cuba.

A major concern for the paper is that there are frequent misspellings that detracts from the message. These are easily addressable but would need to be corrected prior to publication.

The manuscript has been revised by a native English speaker working in the microbiology field.

Minor Concerns

It wasn’t clear what is meant by the statement “Only subgenotype C2 showed significant higher viral loads than subgenotype C3”. Does this mean that C3 and C2 subtype samples were observed to have higher viral loads than the other KSHV K1 subtypes found in the study, for instance A3 and A5, but that C2 displayed the highest VL overall? This is later clarified in the results section, lines 241-245, but as a summary point in the abstract needs clarification.

Abstract has a limited number of words and we decided to state only when differences were found. However, this point has been rewritten to clarify it.

KSHV K1 A5, B1, C3 etc. are more commonly referred to as “subtypes” in the literature. Don’t necessarily need to replace “sub-genotype” description with “subtype”, just an observation.

Subgenotype has been replaced by subtype 

Jary et al. recently defined the F2 subtype as distinct from F subtypes commonly found in Africa, now referred to as F1. This can be added to the introduction in line 87.

reference 17 has been added

In line 147, it was noted that only three A3 subtypes were used for comparison in the sequence alignments. Did BLAST searches of the newly described A3 indicate homology to any other published KSHV K1 subtype references?

Eight A3 subtype sequences have nbeen added to the study, in total 101 references sequences available at Genbank have been studied

In lines 164-166, it was noted that only one reference was used for comparison for the rare KSHV K1 subtypes. For some, like C6, there are few available, but the sequence comparisons would be strengthened with the addition of more references. The E2 reference sited from the Biggar et al paper would be considered an E1 due to the publication of E subtypes from Cuba (Kouri,V et al AIDS 2007 and Virology 2012). It would be of interest to compare the current study sequence to the Cuban sequences, as previously mentioned, in order to confirm or clarify the E subtype found in Spain. In addition, F2 subtype references should be added from Lacoste,V. Virology 2000 or Jary et al. J. Inf. Diseases 2020 publications.

All these referred sequences have been added to the current study

In lines 225-227, there is some confusion about why the author’s feel that subtype C3 is the most prevalent. The sentence “This could be explained by the fact that the number of subgenotypes obtained for genotype A (A1-A5) was higher than in genotype C (C1, C2, C3 and C5)” doesn’t seem to support why C3 is found more frequently.

This point has been addressed

In section 3.3, could the author’s comment if they think that the availability of samples collected year to year could affect the observations made concerning frequency of subtypes observed?

In sections 3.4 and 3.5, some of the observations in certain categories are based upon small sample sizes which should be discussed as a limitation.

A limitation of study concerning small sampling in certain years has been added to the corresponding section

In the discussion, it was noted that the frequency of KSHV detection did not vary significantly north to south in Spain as has been reported in Italy. Studies in Italy, as well as Uganda, have found associations between malaria transmission and KSHV prevalence. While the authors admit that the study wasn’t designed to address this question, perhaps they could comment on malaria incidence as a risk factor for KSHV in Spain as well as list the social factors referred to in lines 343-345.

This point is discussed in discussion section

In the supplemental alignment tables, not sure what is meant by alignment of “problem” samples? The reference used for alignment is a KSHV A2 subtype. Alignment of sequences that are not A2 would have polymorphisms. Suggest “problem” be removed from the titles or discussed within the manuscript for clarification.

This point has been addressed.

No mention of data availability was found in the manuscript. Will the sequences be submitted to GenBank or some other publicly assessable resource?

All sequence data have been uploaded to GenBank. Therefore we obtained 142 accesion numbers corresponding to 142 sequences yielded in the current study.

Concluding comment

Publications of KSHV sequences from an understudied area would be of interest to the KSHV community. As KSHV K1 subtypes reflect ethnic and geographical differences, the diverse distribution of subtypes noted in the current study highlight that urban centers may have several circulating strains which can vary over time as communities change.

6. PLOS authors have the option to publish the peer review history of their article (what does this mean?). If published, this will include your full peer review and any attached files.

Do you want your identity to be public for this peer review? For information about this choice, including consent withdrawal, please see our Privacy Policy.

Reviewer #1: No

---

## [Editor Report · Decision Letter 1]

22 Aug 2022

MOLECULAR EPIDEMIOLOGY OF KAPOSI SARCOMA VIRUS IN SPAIN

PONE-D-22-01291R1

Dear Dr. Tarragó,

We’re pleased to inform you that your manuscript has been judged scientifically suitable for publication and will be formally accepted for publication once it meets all outstanding technical requirements.

Kind regards,

Dirk P. Dittmer, Ph.D.

Academic Editor

PLOS ONE

Additional Editor Comments (optional):

The authors have addressed the reviewer's comments.
---

## [Editor Report · Acceptance letter]

3 Oct 2022

PONE-D-22-01291R1 

Molecular epidemiology of Kaposi sarcoma virus in Spain 

Dear Dr. Tarragó:

I'm pleased to inform you that your manuscript has been deemed suitable for publication in PLOS ONE. Congratulations! Your manuscript is now with our production department. 

Kind regards, 

on behalf of

Dr. Dirk P. Dittmer 

Academic Editor

PLOS ONE